

# Adjusting eye aspect ratio for strong eye blink detection based on facial landmarks

Christine Dewi[1,2], Rung-Ching Chen[1], Xiaoyi Jiang[3] and Hui Yu[4]

[1] Department of Information Management, Chaoyang University of Technology, Taichung, Taiwan, Taiwan
[2] Department of Information Technology, Satya Wacana Christian University, Salatiga, Central Java, Indonesia
[3] Department of Mathematics and Computer Science, University of Münster, Münster, Germany
[4] School of Creative Technologies, University of Portsmouth, Portsmouth, United Kingdom

## ABSTRACT

Blink detection is an important technique in a variety of settings, including facial movement analysis and signal processing. However, automatic blink detection is very challenging because of the blink rate. This research work proposed a real-time method for detecting eye blinks in a video series. Automatic facial landmarks detectors are trained on a real-world dataset and demonstrate exceptional resilience to a wide range of environmental factors, including lighting conditions, face emotions, and head position. For each video frame, the proposed method calculates the facial landmark locations and extracts the vertical distance between the eyelids using the facial landmark positions. Our results show that the recognizable landmarks are sufficiently accurate to determine the degree of eye-opening and closing consistently. The proposed algorithm estimates the facial landmark positions, extracts a single scalar quantity by using Modified Eye Aspect Ratio (Modified EAR) and characterizing the eye closeness in each frame. Finally, blinks are detected by the Modified EAR threshold value and detecting eye blinks as a pattern of EAR values in a short temporal window. According to the results from a typical data set, it is seen that the suggested approach is more efficient than the state-of-the-art technique.

Corresponding authors
Christine Dewi,
s10714904@gm.cyut.edu.tw
Rung-Ching Chen,
crching@cyut.edu.tw

## INTRODUCTION

Eye blinks detection technology is essential and has been applied in different fields such as the intercommunication between disabled people and computers (*Królak & Strumiłło, 2012*), drowsiness detection (*Rahman, Sirshar & Khan, 2016*), the computer vision syndromes (*Al Tawil et al., 2020*; *Drutarovsky & Fogelton, 2015*), anti-spoofing protection in face recognition systems (*Pan et al., 2007*), and cognitive load (*Wilson, 2002*).

According to the literature review, computer vision techniques rely heavily on the driver's facial expression to determine their state of drowsiness. In *Anitha, Mani & Venkata Rao (2020)*, Viola and Jones face detection algorithms are used to train and classify images sequentially. Further, an alarm will sound if the eyes remain closed for a certain time. Other

research proposed a low-cost solution for driver fatigue detection based on micro-sleep patterns. The classification to find whether eye is closed or open is done on the right eye only using SVM and Adaboost (*Fatima et al., 2020*).

*Ghourabi, Ghazouani & Barhoumi (2020)* recommend a reliable method for detecting driver drowsiness by analyzing facial images. In reality, the blinking detection's accuracy may be reduced in particular by the shadows cast by glasses and/or poor lighting. As a result, drowsiness symptoms include yawning and nodding in addition to the frequency of blinking, which is what most existing works focus solely on. For the classification of the driver's state, *Dreisig et al. (2020)* developed and evaluated a feature selection method based on the k-Nearest Neighbor (KNN) algorithm. The best-performing feature sets yield valuable information about the impact of drowsiness on the driver's blinking behavior and head movements. The Driver State Alert Control system expects to detect drowsiness and collision liability associated with strong emotional factors such as head-shoulder inclination, face detection, eye detection, emotion recognition, estimation of eye openness, and blink counts (*Persson et al., 2021*).

Moreover, investigating an individual's eye state in terms of blink time, blink count, and frequency provides valuable information about the subject's mental health. The result uses to explore the effects of external variables on changes in emotional states. Individuals' normal eyesight is characterized by the presence of spontaneous eye blinking at a certain frequency. The following elements impact eye blinking, including the condition of the eyelids, the condition of the eyes, the presence of illness, contact lenses, the psychological state, the surrounding environment, medicines, and other stimuli. The blinking frequency ranges between 6 and 30 times per minute (*Rosenfield, 2011*). Furthermore, the term "eye blink" refers to the quick shutting and reopening of the eyelids, which normally lasts between 100 and 400 ms. Reflex blinking occurs significantly faster than spontaneous blinking, which occurs significantly less frequently. The frequency and length of blinking may be influenced by relative humidity, temperature, light, tiredness, illness, and physical activity. Real-time facial landmark detectors (*Čech et al., 2016*; *Dong et al., 2018*) are available that captures most of the distinguishing features of human facial images, including the corner of the eye angles and eyelids.

A person's eye size does not match another's eye; for example, one has big eyes but the other has small eyes. They don't have the same eyes or height value, as expected. When a person with small eyes closes his or her eyes, he or she may appear to have the same eye height as a person with large eyes. This issue will affect the experimental results. Therefore, we propose a simple but effective technique for detecting eye blink using a newly developed facial landmark detector with a modified Eye Aspect Ratio (EAR). Because our objective is to identify endogenous eye blinks, a typical camera with a frame rate of 25–30 frames per second (fps) is adequate. Eye blinks disclosure can be based on motion tracking within the eye region (*Divjak & Bischof, 2009*).

*Lee, Lee & Park (2010)* try to estimate the state of an eye, including an eye open or closed. *García et al. (2012)* experiment with eye closure for individual frames, which is consequently used in a sequence for blink detection. Other methods compute a difference between frames, including pixels values (*Kurylyak, Lamonaca & Mirabelli, 2012*) and

descriptors (*Malik & Smolka, 2014*). Using the effective Eye Aspect Ratio (*Maior et al., 2020*) and face landmarks (*Mehta et al., 2019*) methods, we developed our own algorithm to perfect it. Another method for blink detection is based on template matching (*Awais, Badruddin & Drieberg, 2013*). The templates with open and/or closed eyes are learned and normalized cross-correlation.

Eye blinks can also be detected by measuring ocular parameters, for example by fitting ellipses to eye pupils (*Bergasa et al., 2006*) using the modification of the algebraic distance algorithm for conic approximation. The frequency, amplitude, and duration of mouth and eye opening and closing play an important role in identifying a driver's drowsiness, according to *Bergasa et al. (2006)*. Adopting EAR as a metric to detect blink in *Rakshita (2018)* yields interesting results in terms of robustness. Based on the previous research result, the blinks rate is previously determined using the EAR threshold value 0.2. Due to the large number of individuals involved and the variation and features between subjects, such as natural eye openness, this approach was considered impractical for this study.

The most significant contributions made by this paper are as follows: (1) Blink types are automatically classified using a method that defines a new threshold based on the Eye Aspect Ratio value as a new parameter called Modified EAR. A detailed description of this method is provided in the paper. (2) Adjusted Eye Aspect Ratio for Strong Blink Detection based on facial landmarks was performed in this experiment. Then, we analyze and discuss in detail the experimental results with public datasets including the Talking Face and Eyeblink8 datasets. (3) We proposed a new Eye Video S1 dataset, and our dataset has the unique characteristics of people with small eyes and glasses. (4) Our experimental results show that using the proposed Modified EAR as a new threshold can improve blink detection results in the experiment.

Furthermore, this research work is organized as follows. Related work and the approach we intend to use in this study describes in the Materials and Methods section. Section 4 describes the experiment and results. A detailed description of the findings of our study is provided in Section 4. Finally, conclusions are drawn, and future work is proposed in Section 5.

## MATERIALS & METHODS

### Eye blink detection with facial landmarks

Eye blinking is a suppressed process that involves the rapid closure and reopening of the eyelid. Multiple muscles are involved in the blinking of the eyes. The orbicularis oculi and levator palpebrae superioris are the two primary muscles that regulate eye closure and opening. Blinking serves some important purposes, one of which is to moisten the corner of an individual's eye. Additionally, it cleans the cornea of the eye when the eyelashes are unable to catch all of the dust and debris that enter the eye. Everyone must blink to spread tears over the entire surface of the eyeball, and especially over the surface of the cornea. Blinking also performs as a reflex to prevent foreign objects from entering the eye. The goal of facial landmark identification is to identify and track significant landmarks on the face. Face tracking becomes strong for rigid facial deformation and not stiff due to

head movements and facial expressions. Furthermore, facial landmarks were successfully applied to face alignment, head pose estimation, face swapping (*Chen et al., 2019*), and blink detection (*Cao et al., 2021*).

*Kim et al. (2020)* implemented semantic segmentation to accurately extract facial landmarks. Semantic segmentation architecture and datasets containing facial images and ground truth pairs are introduced first. Further, they propose that the number of pixels should be more evenly distributed according to the face landmark in order to improve classification performance. *Utaminingrum et al. (2021)* suggested a segmentation and probability calculation for a white pixel analysis based on facial landmarks as one way to detect the initial position of an eye movement. Calculating the difference between the horizontal and vertical lines in the eye area can be used to detect blinking eyes. Another research study (*Navastara, Putra & Fatichah, 2020*) reported that the features of eyes we extracted using a Uniform Local Binary Pattern (ULBP) and the Eyes Aspect Ratio (EAR).

In our research we implement the Dlib's 68 Facial landmark (*Kazemi & Sullivan, 2014*). The Dlib library's pre-trained facial landmark detector is used to estimate 68 (x, y)-coordinates corresponding to facial structures on the face. The 68 coordinates' indices Jaw Points = 0–16, Right Brow Points = 17–21, Left Brow Points = 22–26, Nose Points = 27–35, Right Eye Points = 36–41, Left Eye Points = 42–47, Mouth Points = 48–60, Lips Points = 61–67 and shown in Fig. 1. Facial landmark points identification using *Dlib's* 68 Model consists of the following two steps: (1) Face detection: Face detection is the first method that locates a human face and returns a value in *x, y, w, h* which is a rectangle. (2) Face landmark: After getting the location of a face in an image, we have to through points inside the rectangle. This annotation is part of the 68-point iBUG 300-W dataset on which the *Dlib* face landmark predictor is trained. Whichever data set is chosen, the *Dlib* framework can be used to train form predictors on the input training data.

## Eye Aspect Ratio (EAR)

Eye Aspect Ratio (EAR) is a scalar value that responds, especially for opening and closing eyes (*Sugawara & Nikaido, 2014*). A drowsy detection and accident avoidance system based on the blink duration was developed by *Pandey & Muppalaneni (2021)* and their work system has shown the good accuracy on yawning dataset (YawDD). To distinguish between the open and closed states of the eye, they used an EAR threshold of 0.3. Figure 2 depicts the progression of time it takes to calculate a typical EAR value for one blink. During the flashing process, we can observe that the EAR value increases or decreases rapidly. According to the results of previous studies, we used threshold values to identify the rapid increase or decrease in EAR values caused by blinking. As per previous research, we know that setting the threshold at 0.2 is beneficial for the work at hand. In addition to this approach, many additional approaches to blink detection using image processing techniques have been suggested in the literature. However, they have certain drawbacks, such as strict restrictions on image and text quality, which are difficult to overcome. Based on the previous research result, we selected EAR threshold of 0.2 and 0.3 in our experiment. EAR formula is insensitive to the direction and distance of the face, thus providing the benefit of identifying faces from a distance. EAR value calculates by substituting six

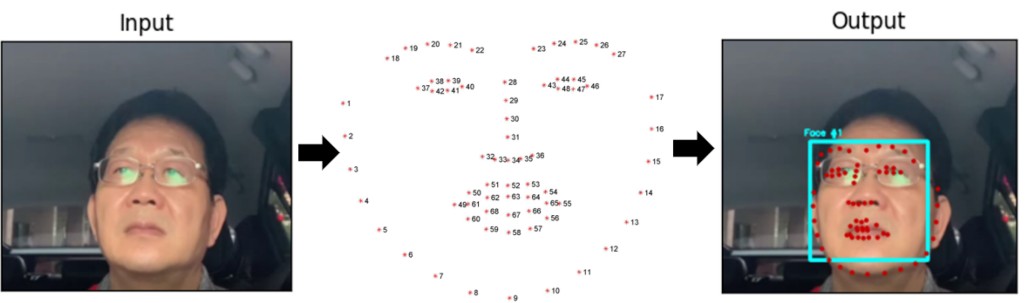

**Figure 1** Eye detection process using facial landmarks (right eye points = 36–41, left eye points = 42–47).

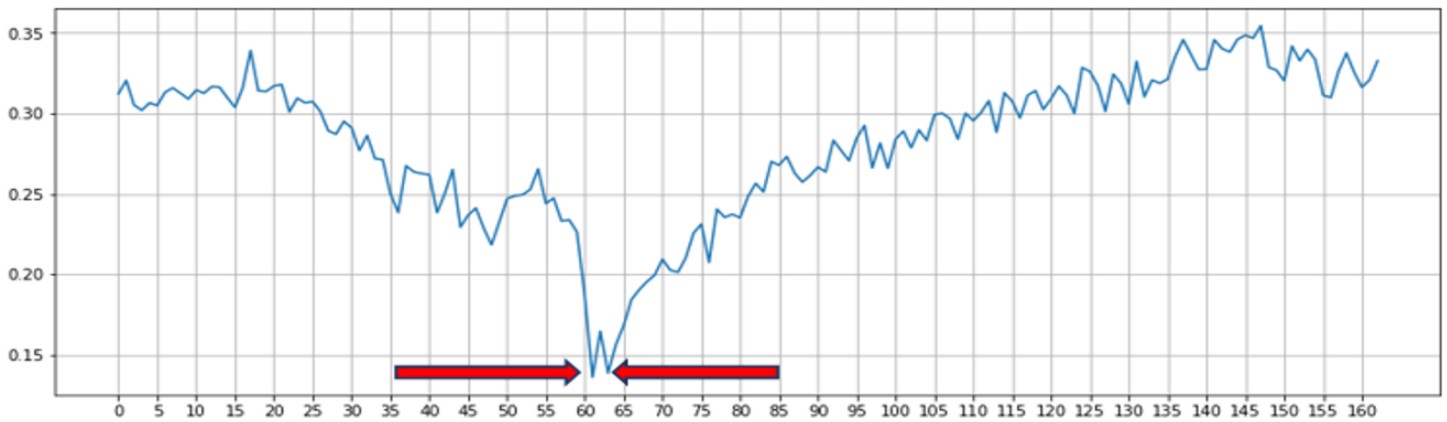

**Figure 2** Single blink detection process and the first blink is detected between the 60th and 65th frames.

coordinates around the eyes shown in Fig. 3 into Eqs. (1) –(2) (*You et al., 2019*; *Noor et al., 2020*).

$$EAR = \frac{\|P2 - P6\| + \|P3 - P5\|}{2\|P1 - P4\|} \tag{1}$$

$$AVG\ EAR = \frac{1}{2}(EAR_{Left} + EAR_{Right}). \tag{2}$$

Equation (1) describes the EAR equations, where P1 to P6 represent the 2D landmark positions on the retina. As illustrated in Fig. 3, P2, P3, P5, and P6 were used to measure eye height, while P1 and P4 were used to measure eye width. When the eyes are opened, the EAR of the eyes remains constant, but when the eyes are closed, the EAR value rapidly decreases to almost zero, as shown in Fig. 3B.

## Modified Eye Aspect Ratio (Modified EAR)

Based on the fact that people have different eye sizes, in this study, we recalculate the EAR (*Huda, Tolle & Utaminingrum, 2020*) value used as a threshold. In this research, we

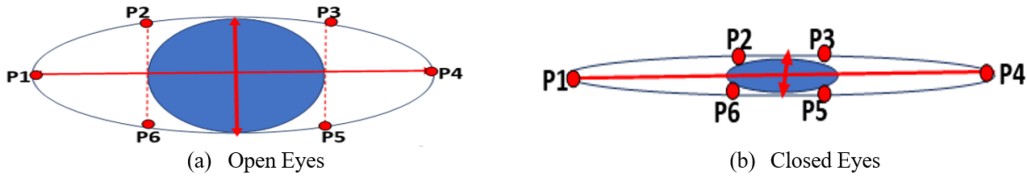

| (a) Open Eyes | (b) Closed Eyes |

**Figure 3  The examples of open eyes and closed eyes with facial landmarks (P1-P6).**

proposed the modified eye aspect ratio (Modified EAR) for closed eyes with Eq. (3) and open eyes with Eq. (4).

$$EAR_{Closed} = \frac{\|P2-P6\|_{min} + \|P3-P5\|_{min}}{2\|P1-P4\|_{max}} \tag{3}$$

$$EAR_{Open} = \frac{\|P2-P6\|_{max} + \|P3-P5\|_{max}}{2\|P1-P4\|_{min}} \tag{4}$$

From Eqs. (3) and (4) we calculate our Modified EAR in Eq. (5)

$$Modified\ EAR_{Threshold} = (EAR_{Open} + EAR_{Closed})/2 \tag{5}$$

*Eye Status*

$$\begin{cases} EAR \leq EAR_{Threshold} = EyeClosed \\ EAR \geq EAR_{Threshold} = EyeOpen \end{cases} \tag{6}$$

Equation (6) depicts the EAR output range while the eyes are open and closed. When the eyes are closed, the EAR value will be close to 0, but the EAR value may be any integer larger than 0 when the eyes are open.

## Eye blink detection flowchart

Figure 4 describes eye blink detection process. The first step is to divide the video into frames. Next, the Facial landmarks feature (*Wu & Ji, 2019*) is implemented with the help of *Dlib* to detect the face. The detector used here is made up of classic Histogram of Oriented Gradients (HOG) (*Dhiraj & Jain, 2019*) feature along with a linear classifier. Facial landmarks detector is implemented inside *Dlib* (*King, 2009*) to detect facial features like eyes, ears, and nose.

Following the detection of the face, the eye area is identified using the facial landmarks dataset. We can identify 68 landmarks (*Yin et al., 2020*) on the face using this dataset. A corresponding index accompanies each landmark. The targeted area of the face is identified *via* the application of the index criteria. Point index for two eyes as follows: (1). Left eye:(37, 38, 39, 40, 41, 42), (2). Right eye: (43, 44, 45, 46, 47, 48) (*Ling et al., 2021*; *Tang et al., 2018*). After extracting the eye region, it is processed for detecting eye blinks. The eye region discovery is made at the beginning stage of the system.

Our research detects the blinks with the help of two lines. Lines are drawn horizontally, and vertically splitting the eye. The act of temporarily closing the eyes and moving the

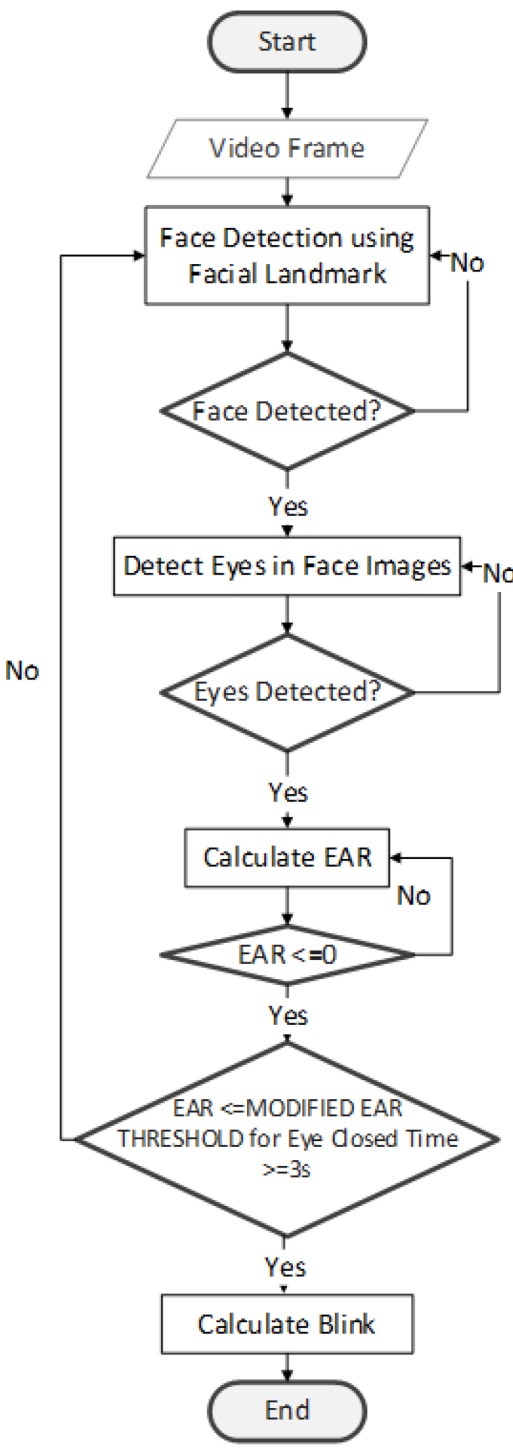

**Figure 4** Eye blink detection flowchart.

eyelids is referred to as blinking. Blinking eyes is a rapid natural process. We can assume that the eye is closed/blinked when: (1) Eyeball is not visible, (2) eyelid is closed, (3) the upper and lower eyelids are connected. For an opened eye, both vertical and horizontal lines are almost identical, while for a closed eye, the vertical line becomes smaller or almost vanished. This research study sets a threshold value based on Modified EAR equations. If the EAR is smaller than the Modified EAR Threshold for 3 s, we can consider eyes blink. In our experiment, we implement three different threshold value 0.2, 0.3, and modified the EAR threshold for each video dataset.

## Eye blink dataset

The Eyeblink8 dataset is more challenging as it includes facial emotions, head gestures, and looking down on a keyboard. This dataset consists of 408 blinks on 70,992 video frames, as annotated by *Fogelton & Benesova (2016)*, with a video resolution of 640 × 480 pixels. The video was captured at 30 fps with an average length from 5,000 to 11,000 frames. The talking face dataset consists of one video recording of one subject talking in front of the camera. The person in the video is making various facial expressions, including smiles, laughing, and funny face. Moreover, this video clip is captured with 30 fps with a resolution of 720 × 576 and contains 61 annotated blinks (*Fogelton & Benesova, 2016*).

The annotations start with line "#start" and rows consist of the following information *frame ID: blink ID: NF: LE_FC: LE_NV: RE_FC: RE_NV: F_X: F_Y: F_W: F_H: LE_LX: LE_LY: LE_RX: LE_RY: RE_LX: RE_LY: RE_RX: RE_RY.* The example of a frame consist a blink as follows: *2851: 9: X: X: X: X: X: 240: 204: 138: 122: 258: 224: 283:225:320:226:347:224.* A blink may consist of fully closed eyes or not. According to blinkmatters.com, the scale of a blink consists of fully closed eyes was 90% to 100%. The row will be like: *2852: 9: X: C: X: C: X: 239: 204: 140: 122: 259: 225: 284: 226: 320: 227: 346: 226.* We are only interested in the blink ID and eye completely closed (FC) columns in our study experiment, therefore, we will ignore any other information.

TalkingFace dataset consists of one video recording of one subject talking in front of the camera and making different facial expressions. This video clip is captured with 25 fps with a resolution of 720 × 576 and contains 61 annotated blinks (*Drutarovsky & Fogelton, 2015*; *Fogelton & Benesova, 2016*).

Modified EAR threshold equation implemented for each dataset. After calculation, the EAR threshold for the Talking face dataset is 0.2468, Eyeblink Video S4 0.2923, Eyeblink Video S8 0.2105, and 0.2103 for Eye Video S1. The dataset information explains in Table 1.

Eye Video S1 dataset labeling procedure using Eyeblink annotator 3.0 by *Fogelton & Benesova (2016)*. The annotation tool uses OpenCV 2.4.6. Eye Video S1 were captured at 29.97 fps and has a length of 1829 frames with 29.6 MB. Our dataset has the unique characteristics of people with small eyes and glasses. The environment is the people who drive the car. This dataset can be used for further research. It is difficult to find a dataset of people with small eyes, wearing glasses, and driving cars based on our knowledge. We collect the video from the car dashboard camera in Wufeng District, Taichung, Taiwan.

**Table 1  Dataset Information.**

| Variable | Description |
| --- | --- |
| frame ID | In a separate file, a frame counter may be used to get a timestamp. |
| blink ID | A unique blink ID is defined as a series of identical blink ID frames. An eye blink interval is defined as a sequence of identical blink ID frames. |
| non frontal face (NF) | While the individual is gazing to the side and blinking, the supplied variable changes from X to N. |
| left eye (LE), | Left eye. |
| right eye (RE), | Right eye. |
| face (F) | Face. |
| eye fully closed (FC) | If the subject's eyes are closed between 90% and 100%, the provided flag will change from X to C. |
| eye not visible (NV) | While the subject's eye is obscured by the hand, poor lighting, or even excessive head movement, this variable shifts from X to N. |
| face bounding box (F_X, F_Y, F_W, F_H) | x and y coordinates, width, height. |
| left and right eye corners positions | RX (right corner x coordinate), LY (left corner y coordinate) |

# RESULTS

Tables 2 and 3 explains the statistics on the prediction and test set for each video dataset. Talking Face dataset capture with 30 fps, 5,000 frames, and the duration is 1,667.67 s. Statistics on the prediction set show that the number of closed frames processed is 292, and the number of blinks is 42 for EAR threshold 0.2. However, statistics on the test set describe the number of closed frames as 153, and the number of blinks is 61. This experiment exhibits an accuracy of 96.85% and an AUC of 94.68%. The highest AUC score for the Talking Face dataset was achieved while using Modified EAR threshold 0.2468; it obtains 96.85%.

Furthermore, Eyeblink8 dataset Video S4 processed 5,454 frames with 30 fps and duration of 181.8 s. The maximum AUC obtains while implementing our Modified EAR threshold was 0.2923; it achieved 91.17%. Moreover, Eyeblink8 dataset Video S8 contains 10,712 frames with 30 fps and durations 357.07 s. This dataset also got the best AUC when employed Modified EAR threshold 0.2105, it achieves 96.60%. Eyeblink8 dataset Video S8 exhibits the minimum result of 21.1% accuracy and 60.2% AUC while employed EAR threshold 0.2. In this study, we prioritize AUC because of some reasons. First, the AUC is scale-invariant and assesses how well the predictions are ordered rather than how well they are ordered in real numbers. Second, AUC is not affected by categorization limits. It assesses the prediction accuracy of the models, regardless of the categorization criteria used to assess them.

The talking Face video dataset exhibits 94% accuracy and 96.85% AUC. Followed by Eyeblink8, Video S8 achieves 95% accuracy and 96% AUC. Further Eyeblink8 Video S4

**Table 2  Statistics on prediction and test set on the talking Face and Eyeblink8 datasets.**

| Dataset | Talking Face | | | Eyeblink8 Video 4 | | | Eyeblink8 Video 8 | | |
|---|---|---|---|---|---|---|---|---|---|
| Video Info | | | | | | | | | |
| FPS | 30 | 30 | 30 | 30 | 30 | 30 | 30 | 30 | 30 |
| Frame Count | 5000 | 5000 | 5000 | 5454 | 5454 | 5454 | 10712 | 10712 | 10712 |
| Durations (s) | 166.67 | 166.67 | 166.67 | 181.8 | 181.8 | 181.8 | 357.07 | 357.07 | 357.07 |
| EAR Threshold (t) | 0.2 | 0.3 | 0.2468 | 0.2 | 0.3 | 0.2923 | 0.2 | 0.3 | 0.2105 |
| Statistics on the prediction set are | | | | | | | | | |
| Total Number of Frames Processed | 5000 | 5000 | 5000 | 5315 | 5315 | 5315 | 10663 | 10663 | 10663 |
| Number of Closed Frames | 292 | 1059 | 458 | 123 | 1081 | 1035 | 8520 | 1991 | 628 |
| Number of Blinks | 42 | 78 | 58 | 15 | 54 | 54 | 347 | 124 | 51 |
| Statistics on the test set are | | | | | | | | | |
| Total Number of Frames Processed | 5000 | 5000 | 5000 | 5315 | 5315 | 5315 | 10663 | 10663 | 10663 |
| Number of Closed Frames | 153 | 153 | 153 | 117 | 117 | 117 | 107 | 107 | 107 |
| Number of Blinks | 61 | 61 | 61 | 31 | 31 | 31 | 30 | 30 | 30 |
| Eye Closeness Frame by Frame Test Scores | | | | | | | | | |
| Accuracy | 0.9678 | 0.819 | 0.94 | 0.982 | 0.819 | 0.8273 | 0.211 | 0.8233 | 0.951 |
| AUC | 0.9486 | 0.907 | 0.9685 | 0.803 | 0.907 | 0.9117 | 0.602 | 0.9108 | 0.966 |

**Table 3  Statistics on prediction and test set on the Eye Video 1 dataset.**

| Dataset | Eye Video 1 | | |
|---|---|---|---|
| Video Info | | | |
| FPS | 29.97 | 29.97 | 29.97 |
| Frame Count | 1829 | 1829 | 1829 |
| Durations (s) | 61.03 | 61.03 | 61.03 |
| EAR Threshold (t) | 0.2 | 0.3 | 0.2103 |
| Statistics on the prediction set are | | | |
| Total Number of Frames Processed | 1829 | 1829 | 1829 |
| Number of Closed Frames | 77 | 888 | 56 |
| Number of Blinks | 7 | 93 | 6 |
| Statistics on the test set are | | | |
| Total Number of Frames Processed | 1829 | 1829 | 1829 |
| Number of Closed Frames | 58 | 58 | 58 |
| Number of Blinks | 14 | 14 | 14 |
| Eye Closeness Frame by Frame Test Scores | | | |
| Accuracy | 0.9273 | 0.5014 | 0.9388 |
| AUC | 0.4872 | 0.4006 | 0.4931 |

obtains 83% accuracy and 91.17% AUC. Although the Talking Face dataset gets a high accuracy of 97% when using an EAR threshold of 0.2, it reaches the lowest AUC of 94.86%.

In Table 3, Eye Video S1 dataset processed 1829 frames with 29.97 fps and duration of 61.03 s. The maximum AUC 0.4931% and accuracy 93.88% were obtains while implementing our Modified EAR threshold 0.2103.

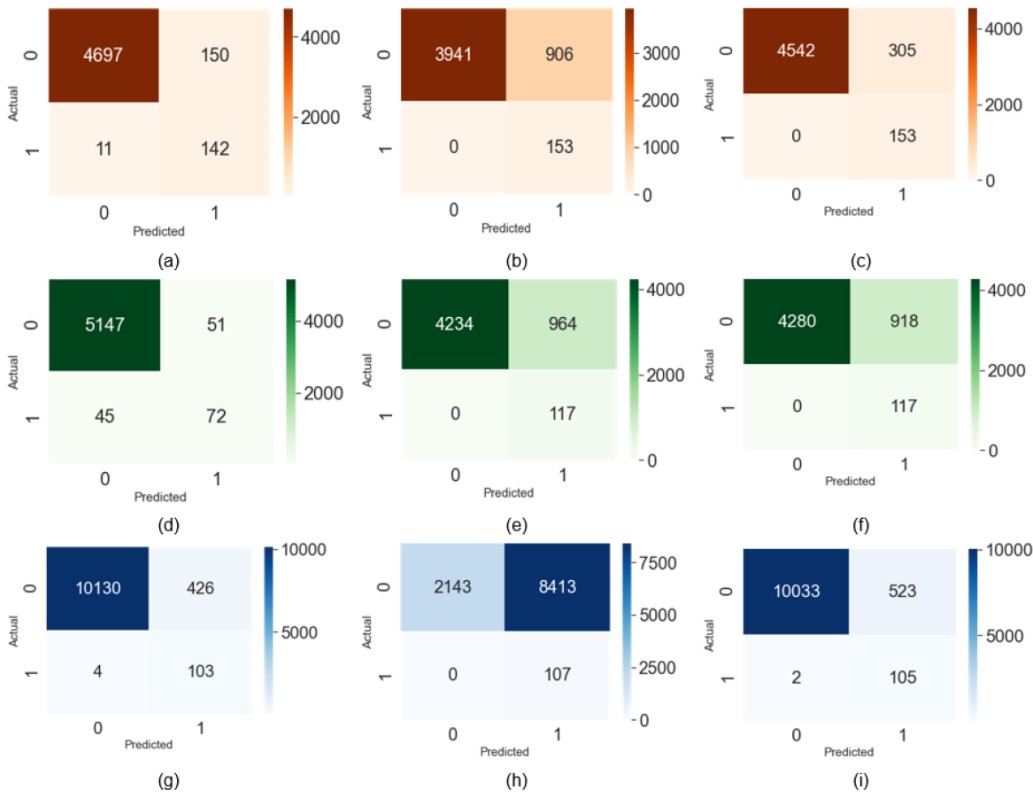

**Figure 5** Confusion matrix on the talking face dataset (true positive (TP), false positive (FP), true negative (TN), false negative (FN)).

Figures 5A–5C shows the confusion matrix for the Talking Face dataset. Figures 5D–5F explains the confusion matrix for the Eyeblink8 dataset Video S4. The confusion matrix for Eyeblink8 dataset Video S8 is drawn in Figs. 5G-5I. Figure 5I describes the false positive (FP) 2 out of 107 positive labels (0.0187%) and false negative (FN) 523 out of 10556 negative labels (0.0495%).

Tables 4 and 5 represents the evaluation performance of Precision, Recall, and F1 Score for each dataset in detail. The experimental results show that our proposed method, Modified EAR has the best performance compared to others. Furthermore, researchers have only used 0.2 or 0.3 as the EAR threshold, even though not all people's eye sizes are the same. Therefore, it is better to recalculate the EAR threshold to determine whether the eye is closed or open to identify the blink more precisely.

Based on Tables 4 and 5 we can conclude that, when we recalculate the EAR threshold value the experiment achieves the best performance for all datasets. The TalkingFace dataset applies an EAR threshold of 0.2468, Eyeblink8 Video S4 uses an EAR threshold of 0.2923, Eyeblink8 Video S8 uses 0.2105 as an EAR threshold, and Eye Video S1 uses 0.2103 as an EAR threshold value. Our research experiment processes video frame by frame and detects eye blink for every three frames shown in Fig. 6. The experiment results just show the beginning, middle, and finish frames of blinks. Figure 6 illustrates the Eye Video S1

Dewi et al. (2022), *PeerJ Comput. Sci.*, DOI 10.7717/peerj-cs.943

**Table 4   Evaluation performance of precision, recall, and F1-score on the talking face and Eyeblink8 datasets.**

| Evaluation | Talking face | | | | Eyeblink8 Video 4 | | | | Eyeblink8 Video 8 | | | |
|---|---|---|---|---|---|---|---|---|---|---|---|---|
| | Precision | Recall | F1-score | Support | Precision | Recall | F1-score | Support | Precision | Recall | F1-score | Support |
| | EAR Threshold (t) = 0.2 | | | | EAR Threshold (t) = 0.2 | | | | EAR Threshold (t) = 0.2 | | | |
| 0 | 0.99 | 0.97 | 0.98 | 4847 | 0.99. | 0.99 | 0.99 | 5198 | 0.99 | 0.20 | 0.34 | 10556 |
| 1 | 0.49 | 0.93 | 0.64 | 153 | 0.59 | 0.62 | 0.60 | 117 | 0.01 | 1.00 | 0.02 | 107 |
| Macro avg | 0.74 | 0.95 | 0.81 | 5000 | 0.79 | 0.80 | 0.80 | 5315 | 0.51 | 0.60 | 0.18 | 10663 |
| Weight avg | 0.98 | 0.97 | 0.97 | 5000 | 0.98 | 0.98 | 0.98 | 5315 | 0.99 | 0.21 | 0.33 | 10663 |
| Accuracy | | 0.97 | | 5000 | | | 0.98 | 5315 | | | 0.21 | 10663 |
| | EAR Threshold (t) = 0.3 | | | | EAR Threshold (t) = 0.3 | | | | EAR Threshold (t) = 0.3 | | | |
| 0 | 0.99 | 0.81 | 0.9 | 4847 | 0.99 | 0.81 | 0.9 | 5198 | 0.99 | 0.82 | 0.90 | 10556 |
| 1 | 0.14 | 1.00 | 0.25 | 153 | 0.11 | 1.00 | 0.2 | 117 | 0.05 | 1.00 | 0.10 | 107 |
| Macro avg | 0.57 | 0.91 | 0.57 | 5000 | 0.55 | 0.91 | 0.55 | 5315 | 0.53 | 0.91 | 0.50 | 10663 |
| Weight avg | 0.97 | 0.82 | 0.88 | 5000 | 0.98 | 0.82 | 0.88 | 5315 | 0.99 | 0.82 | 0.89 | 10663 |
| Accuracy | | 0.82 | | 5000 | | | 0.82 | 5315 | | | 0.82 | 10663 |
| | **EAR Threshold (t) = 0.2468** | | | | **EAR Threshold (t) = 0.2923** | | | | **EAR Threshold (t) = 0.2105** | | | |
| 0 | 0.99 | 0.94 | 0.97 | 4847 | 0.99 | 0.82 | 0.90 | 5198 | 0.99 | 0.95 | 0.97 | 10556 |
| 1 | 0.33 | 1.00 | 0.50 | 153 | 0.11 | 1.00 | 0.20 | 117 | 0.17 | 0.97 | 0.27 | 107 |
| Macro avg | 0.67 | 0.97 | 0.73 | 5000 | 0.56 | 0.91 | 0.55 | 5315 | 0.58 | 0.98 | 0.63 | 10663 |
| Weight avg | 0.98 | 0.94 | 0.95 | 5000 | 0.98 | 0.83 | 0.89 | 5315 | 0.99 | 0.95 | 0.97 | 10663 |
| Accuracy | | 0.94 | | 5000 | | | 0.83 | 5315 | | | 0.95 | 10663 |

**Table 5  Evaluation performance of precision, recall, and F1-score on the eye video 1 Dataset.**

| Evaluation | Eye Video 1 | | | |
|---|---|---|---|---|
| | Precision | Recall | F1-score | Support |
| | EAR Threshold (t) = 0.2 | | | |
| 0 | 0.97 | 0.96 | 0.96 | 1771 |
| 1 | 0.01 | 0.02 | 0.01 | 58 |
| Macro avg | 0.49 | 0.49 | 0.49 | 1829 |
| Weight avg | 0.94 | 0.93 | 0.93 | 1829 |
| Accuracy | | | 0.93 | 1829 |
| | EAR Threshold (t) = 0.3 | | | |
| 0 | 0.96 | 0.51 | 0.66 | 1771 |
| 1 | 0.02 | 0.29 | 0.04 | 58 |
| Macro avg | 0.49 | 0.4 | 0.35 | 1829 |
| Weight avg | 0.93 | 0.5 | 0.64 | 1829 |
| Accuracy | | | 0.5 | 1829 |
| | **EAR Threshold (t) = 0.2103** | | | |
| 0 | 0.97 | 0.97 | 0.97 | 1771 |
| 1 | 0.02 | 0.02 | 0.02 | 58 |
| Macro avg | 0.49 | 0.49 | 0.49 | 1829 |
| Weight avg | 0.94 | 0.94 | 0.94 | 1829 |
| Accuracy | | | 0.94 | 1829 |

dataset result. The first blink started at: 1th frame, middle of action at: 5th frame, ended at: 8th frame. Moreover, the second blink started at: 224th frame, middle of action at: 227th frame, ended at: 229th frame.

## DISCUSSION

In our research work, the AUC score is more important than accuracy because of several reasons as follow *Lobo, Jiménez-valverde & Real (2008)*: (1) Our experiment is concerned with ranking predictions, not with producing well-calibrated probabilities. (2) The video dataset is heavily imbalanced. It was discussed extensively in the research paper by *Saito & Rehmsmeier (2015)*. The intuition is the following: the false-positive rate for highly imbalanced datasets is pulled down due to many true negatives. (3) We concentrated on classes that were both positive and negative. If we are as concerned with true negatives as we are with true positives, it makes sense to utilize AUC.

Figure 7 describes the EAR and Error analysis of the Talking Face video dataset. Consider that the optimal slope of the linear regression is m >= 0. Our experiment plots the whole data and obtains $m = 0$. Blinking is infrequent and has a negligible effect on the overall EAR measurement trend. Moreover, Cumulative error is meaningless for blinks due to its delayed impact. Additionally, errors perform more like properly distributed data than the EAR values explain in Fig. 7.

In this paper, the performance of the proposed eye blink detection technique is evaluated by comparing detected blinks with ground-truth blinks using the two standard datasets described above. The output examples can be classified into three classes. True positive

```
1st blink started at: 1st frame, middle of action at: 5th frame, ended
at: 8th frame
```

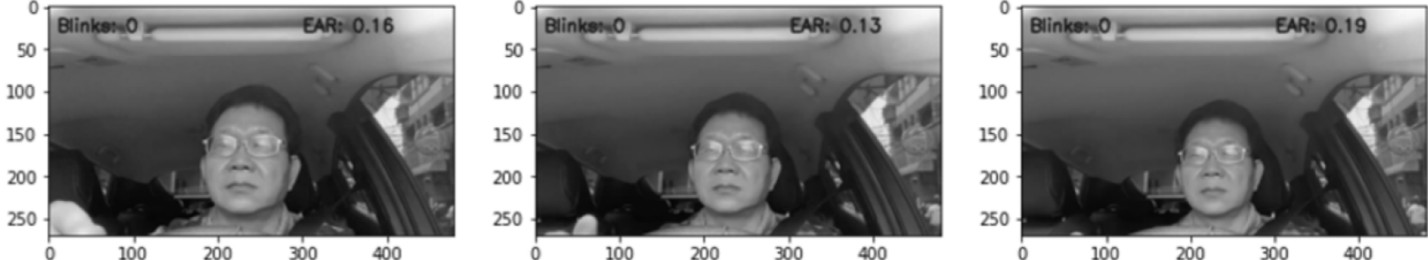

```
2nd blink started at: 224th frame, middle of action at: 227th frame,
ended at: 229th frame
```

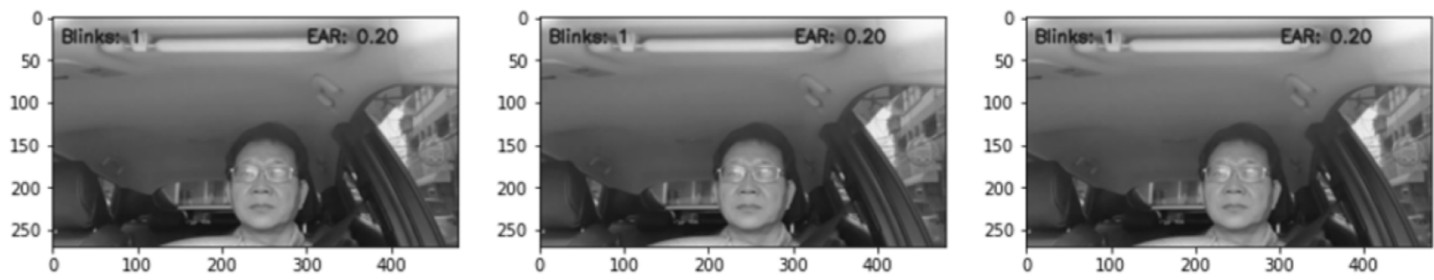

**Figure 6** Eye **Video S1** dataset result. First blink started at: 1st frame, middle of action at: 5th frame, ended at: 8th frame.

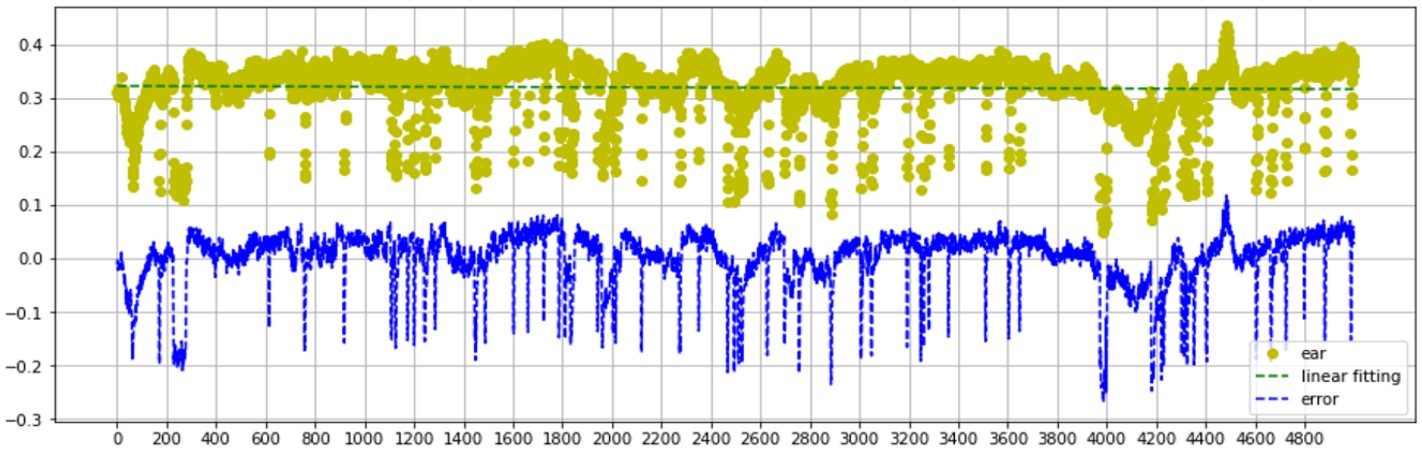

**Figure 7** Talking face dataset EAR and error analysis range between 0–5000 frames.

(TP) is the number of correctly recognized samples; false positive (FP), which assigns to the number of samples with correct identification; false negative (FN), which assigns to the number of samples with incorrect identification; true negative (TN) is the number of unrecognized samples. Precision and recall are represented by *Dewi et al. (2021)*, *Yang et*

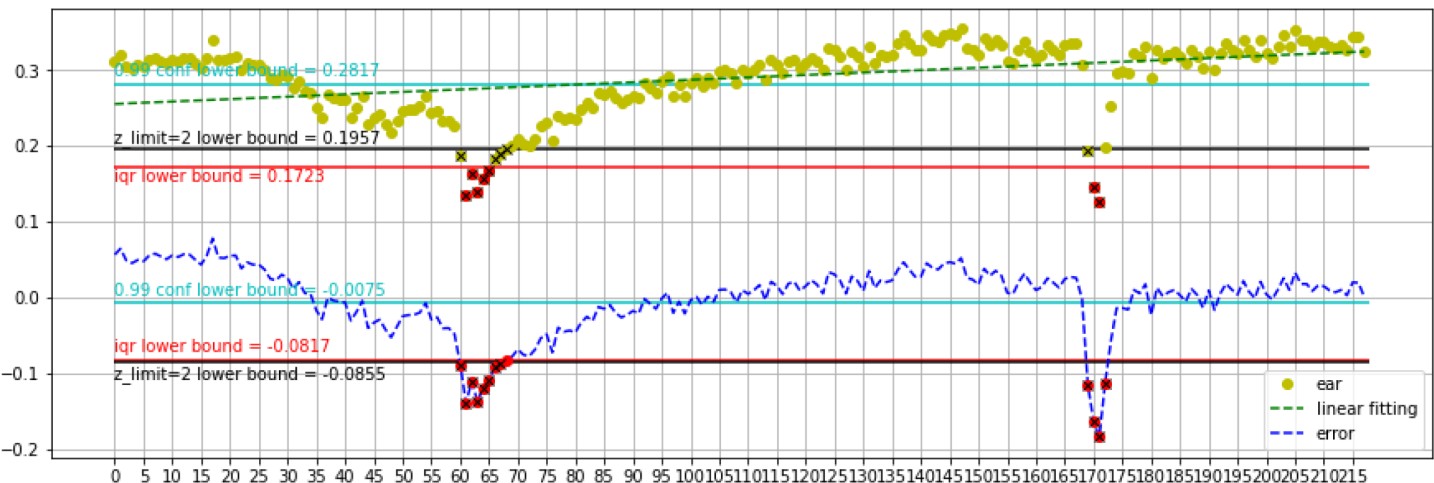

**Figure 8** First blink and second blink talking face dataset analysis in range 0–215 frames.

*al., (2019)* and *Dewi et al. (2021)* in Eqs. (7)–(9).

$$Precision(P) = \frac{TP}{TP + FP} \tag{7}$$

$$Recall(R) = \frac{TP}{TP + FN} \tag{8}$$

Another evaluation index, F1 (*Tian et al., 2019*; *Chen et al., 2020*) is shown in Eq. (9).

$$F1 = \frac{2 \times Precision \times Recall}{Precision + Recall} \tag{9}$$

Figure 8 represents the first blink and second blink analysis for the Talking Face video Dataset. The lower bound is better for estimating blinks. The lower bound of calibration show 0.1723 for ear value. Detects 6 frames in details as follows: 4 frames for first blink and 2 frames for second blink. Furthermore, the lower bound of errors obtains −0.0817 error value. Detects 12 frames with 8 frames for the first blink and 4 frames for second blink. Also, analyzing with $z\_limit = 2$ on calibration does better than running on errors. The lower bound of calibration obtains a 0.1723 EAR value. It detects 12 frames with 8 frames for the first blink, 4 frames for the second blink. The lower bound of errors exhibits −0.0817 error value. It detects 12 frames with 8 frames for the first blink, 4 frames for the second blink. Using the mentioned dataset, the proposed method outperforms methods in previous research. The statistics are listed in Table 6. We obtain the highest Precision, 99%, for all datasets. Moreover, our proposed method achieves 97% Precision on Eye Video S1 dataset.

## CONCLUSIONS

This paper proposes a method to automatically classify blink types by determining the new threshold based on the Eye Aspect Ratio value as a new parameter called Modified

**Table 6** Comparison of previous research with the proposed method.

| Reference | Dataset | Precision (%) |
|---|---|---|
| *Lee, Lee & Park (2010)* | Talking Face | 83.30 |
| *Drutarovsky & Fogelton (2015)* | Talking Face | 92.20 |
| *Fogelton & Benesova (2016)* | Talking Face | 95.00 |
| Proposed Method | Talking Face | 98.00 |
| *Drutarovsky & Fogelton (2015)* | Eyeblink8 | 79.00 |
| *Fogelton & Benesova (2016)* | Eyeblink8 | 94.69 |
| *Al-Gawwam & Benaissa (2018)* | Eyeblink8 | 96.65 |
| Proposed Method | Eyeblink8 | 99.00 |
| Proposed Method | Eye Video 1 | 97.00 |

EAR. Adjusted Eye Aspect Ratio for strong eye blink detection based on facial landmarks. We analyzed and discussed in detail the experiment result with the public dataset and our dataset Eye Video S1. Our work proves that using Modified EAR as a new threshold can improve blink detection results.

In the future, we will focus on the dataset that has facial actions, including smiling and yawning. Both basic and adaptive models lack facial emotions such as smiling and yawning. Machine learning methods may be a viable option, and we will implement SVM in our future research.

### Funding

This paper was supported by the Ministry of Science and Technology, Taiwan (MOST-110-2927-I-324-50, MOST- 110-2221-E-324 -010, MOST-109-2622-E-324 -004) and the EU Horizon 2020 program RISE Project ULTRACEPT under Grant 778062. There was no additional external funding received for this study. The funders had no role in study design, data collection and analysis, decision to publish, or preparation of the manuscript.

### Grant Disclosures

The following grant information was disclosed by the authors:
The Ministry of Science and Technology, Taiwan: MOST-110-2927-I-324-50, MOST-110-2221-E-324-010, MOST-109-2622-E-324-004.
The EU Horizon 2020 program RISE Project ULTRACEPT: 778062.

### Competing Interests

The authors declare there are no competing interests.

### Author Contributions

- Christine Dewi conceived and designed the experiments, performed the experiments, analyzed the data, performed the computation work, prepared figures and/or tables, authored or reviewed drafts of the paper, and approved the final draft.

- Rung-Ching Chen conceived and designed the experiments, performed the experiments, analyzed the data, authored or reviewed drafts of the paper, and approved the final draft.
- Xiaoyi Jiang and Hui Yu performed the experiments, authored or reviewed drafts of the paper, and approved the final draft.

### Data Availability

The data is available at GitHub: https://github.com/ChristineDewi/Blinks-Detection.

### Supplemental Information

Supplemental information for this article can be found online at http://dx.doi.org/10.7717/peerj-cs.943#supplemental-information.

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
