# Peer review of "Adjusting eye aspect ratio for strong eye blink detection based on facial landmarks"

_PeerJ Computer Science, doi:10.7717/peerj-cs.943_

## Round 0.1 · original submission · Major Revisions

1) Captions of figures and title of tables are not well written. Please include more description with the purpose of bringing out the gist of the figures and tables to deliver the main message.

2) The gap of knowledge that author wish to fill must be rewritten to include more background and details so that the novelty of the research can be emphasized.

3) Please include at least 10 more recent references (recent 3 years preferably). Please enrich your literature review and revise the literature review to better explain the state of the art instead of just listing out relevant works. Try your best to bridge previous relevant works to your research of this paper clearly.

4) Suggest to add experiments regarding the environmental conditions when taking the images.

5) More explanation of the selection of threshold is required.

·

Basic reporting

interesting topic and the presented writing is adequate. However, majority of the literature references are too old (more than 5 years). The gap of knowledge of the proposed topic is unclear. Too short prior arts have been discussed, thus, not much of gap of knowledge can be identified from the present works. The landmark used in the study are rely on prior reported works from Dlib, and dataset utilized the publicly available resources from eyeblink8. Suggest the author to test on their own dataset.

Experimental design

the proposed methods are inline with their problem statement. Again, I don't see much of novelty of their proposed works here although the methodology are explained sufficiently.

Validity of the findings

Both qualitative and quantitative analyses are adequate.

Additional comments

NA

Reviewer 2 ·

Basic reporting

1. A good motivational sentence is needed in the Introduction section of the study.

2. The problem space of the study should be expressed more clearly. With this explanation, it will be clearer how eye blink detection handles problem solving in the study.

3. Perhaps, the literature section can be improved by mentioning that there are different fields of study related to eyeblink.

Experimental design

1. It should be stated more clearly how the process is followed for the selection of the threshold. If this value is determined manually, how this value is selected should be explained.

2. It has been stated that environmental conditions are important in taking camera images. However, there is no experimental design related to this.

Validity of the findings

well prepared.

Additional comments

I think the study will get better after these edits and corrections.

---

## Round 0.2 · Major Revisions

The previous comments (as follows) were not addressed sufficiently. Please add more detail to these areas.

1) Captions of figures and title of tables are not well written. Please include more description with the purpose of bringing out the gist of the figures and tables to deliver the main message.

2) The gap of knowledge that author wish to fill must be rewritten to include more background and details so that the novelty of the research can be emphasized.

3) Please include at least 10 more recent references (recent 3 years preferably). Please enrich your literature review and revise the literature review to better explain the state of the art instead of just listing out relevant works. Try your best to bridge previous relevant works to your research of this paper clearly.

4) Suggest to add experiments regarding the environmental conditions when taking the images.

5) More explanation of the selection of threshold is required.

Reviewer 2 ·

Basic reporting

The authors performed the necessary corrections and arrangements with a good effort. I think it can be published as it is.

Experimental design

.

Validity of the findings

.

Additional comments

.

---

## Round 0.3 · accepted · Accept

All concerns have been addressed. Congratulations.